

# An Algorithm for Automatic Fitting and Formula Assignment in Atmospheric Mass Spectra

Valter Mickwitz[1], Otso Peräkylä[1], Frans Graeffe[1], Douglas Worsnop[1,2], and Mikael Ehn[1]

[1]Institute for Atmospheric and Earth System Research, University of Helsinki, Helsinki 00014, Finland.
[2]Aerodyne Research Inc., 45 Manning Road, Billerica, Massachusetts 01821, USA.

**Correspondence:** Valter Mickwitz (valter.mickwitz@helsinki.fi) and Mikael Ehn (mikael.ehn@helsinki.fi)

**Abstract.** Mass spectrometry is an established method for studying the chemical composition of gases and particles in the atmosphere. Using this technique, signals corresponding to thousands, or even tens of thousands of compounds may be detected from ambient air. The process of identifying all the peaks in the mass spectra is often arduous and time–consuming, in particular when multiple overlapping peaks are present. This manual peak fitting and identification may take even experienced analysts anywhere from weeks to months to complete, depending on the desired accuracy and completeness.

In this work, we attempted to automate the fitting and formula assignment workflow and evaluate how far the process can get using a "one button" algorithm. The algorithm constructed in this work takes in commonly known parameters specific to the instrument type and by pressing one button, it runs and ultimately provides a list of likely peaks for the mass spectrum. The algorithm utilizes weighted least squares fitting and a modified version of the Bayesian information criterion along with an iterative formula assignment process. We applied it to synthetic mass spectra and both a gas-phase chemical ionization mass spectrometer (CIMS) dataset and an aerosol mass spectrometer (AMS) dataset. The results were largely comparable with manual peak fitting and identification done previously, but were achieved in a fraction of the time. Erroneous assignments mainly appeared at low–intensity signals, with interference from nearby higher intensity signals, a case that is challenging also for manual peak fitting. This algorithm provides an excellent starting point for a peak list, which, if needed, can be manually revised.

The main result of this study is the algorithm itself. While further improvements and tweaks are possible, the algorithm presented here is currently being implemented into the commonly used Tofware analysis software package, to allow easy utilization by the broader community. We hope this can save valuable time of researchers for data interpretation rather than data processing and curation.

## 1 Introduction

Volatile Organic Compounds (VOCs), emitted into the atmosphere from a multitude of activities and processes, both anthro-pogenic and biogenic (Fowler et al., 2009; Goldstein and Galbally, 2007), are key components of atmospheric chemistry. These compounds are oxidized in the atmosphere, forming a vast number of different species, some with low enough volatility to contribute to aerosol formation (Jokinen et al., 2015; Ehn et al., 2014; Zhang et al., 2012; Riipinen et al., 2011; Kulmala



et al., 1998). Understanding the dynamics and impacts of these trace gasses and particles, on both health and climate requires knowledge about their chemical composition and the processes that form and transform them (Masson-Delmotte et al., 2021; Shrivastava et al., 2017; Heal et al., 2012).

To study these compounds and their chemistry, whether it is in the gas or particle phase, mass spectrometers are commonly used (Zhang et al., 2023; Huey, 2007). There is a vast array of variations of different instruments, targeted at different classes
of compounds. These instruments utilize various ionization methods (Riva et al., 2019b; Rissanen et al., 2019; Lopez-Hilfiker et al., 2019; Canagaratna et al., 2007), inlets (Häkkinen et al., 2023; Eichler et al., 2015; Lopez-Hilfiker et al., 2014) and mass analyzers (Boesl, 2017; Batey, 2014; Hu et al., 2005). In all cases peak fitting to the mass spectra is required to be able to identify all the compounds of interest. The Orbitrap mass analyzer exhibits around an order of magnitude better resolving power than Time–of–Flight (ToF) mass analyzers, and can in most cases unambiguously separate all ions encountered in the
mass spectrum (Riva et al., 2019a; Zuth et al., 2018), but ToFs are far more common in the field of atmospheric science. ToFs often have a large number of partially overlapping signals that require careful peak fitting to separate (Stark et al., 2015; Junninen et al., 2010).

Peak fitting and identification are therefore necessary steps in the analysis of data pertaining to the chemistry of the atmosphere. This process can be arduous for analysts, potentially requiring experts with understanding and intuition of the chemical
properties of the studied system, going through each signal in the spectrum individually. Depending on the desired accuracy of the analysis, this process may take researchers from several days up to months to complete for a newly acquired mass spectral dataset. Thus, there have been several attempts and discussion about ways to facilitate the analysis process (Sandström et al., 2024; Alton et al., 2023; Zhang et al., 2019; Stark et al., 2015). However, most of these studies have not focused on automating the peak list generation process, but rather on improving complementary techniques. Stark et al. (2015) did present an algo-
rithm for automated peak list generation, but the work was focused on obtaining bulk chemical properties of the dataset, rather than accurate individual fits. In this work we present an attempt at a fully automatic peak assignment process, to establish a baseline for how accurate such an approach can be made. The ultimate aim is to be able to provide an algorithm that can dramatically decrease the time analysts need to spend on data processing in the form of peak fitting and formula assignment.

We will here describe our "one button" algorithm which, given a number of inputs (mass calibrated spectrum, resolution and
peak-shape functions, and restrictions on the type of ions to be expected from the instrument), provides the user with a list of chemical formulas that are likely to be present in the sample. We describe the working principles of the algorithm in detail, and apply it to both real and synthetic datasets to understand and evaluate its usefulness.

## 2  Methods

This section outlines the methods used to design the algorithm and also includes brief descriptions of the testing datasets. Note
that when discussing these methods, a single charge is assumed for the signals, and therefore *mass* and *mass–to–charge ratio* are used interchangeably. When discussing peaks, the term *position* is used to describe the mass–to–charge ratio where the signal distribution is centered. This is not necessarily the same as the mean, or the peak of that distribution but depends entirely



on the definition of the peak shape function. The peak shape function is a function that describes the shape of the expected
signal from a single type of ion. The algorithm requires four inputs: a mass calibrated mass spectrum, a resolution function, a
peak shape function, and a list of potential formulas. The former three are standard concepts in high resolution peak fitting and
will not be discussed in more detail here (Stark et al., 2015; Junninen et al., 2010). The list of potential formulas was generated
specifically for this work and the method is described in Sect. 2.2.4. In addition to the four necessary inputs, an optional input
of baseline may be provided. A number of algorithm parameters mentioned in later sections may also be tweaked but default
values are used for all datasets described here, unless mentioned otherwise.

## 2.1  Algorithm structure

The algorithm can roughly be divided into 2 parts. The first, or the free fitting part, provides data and initial guesses for where
peaks may be for the following part. The second part, or the peak assignment part, iteratively assigns formulas to the fits from
the free fitting part, and updates the free fit after every formula assigned.

In more detail, the free fitting section of the algorithm simply fits between zero and $n_{max}$ peaks at each unit mass. At this
point there is still no decision made about the number of peaks, so the algorithm starts by fitting 0 peaks, then moves on to
1 peak, and so on. The previous fit is used to initialize the next fit. The data obtained from these fits are later used in determining
the number of peaks to fit at each unit mass, and to initialize the peak assignment part of the algorithm. $n_{max}$ is chosen to be
higher than the highest number of peaks that could realistically be identified at any unit mass. For the tests and gas phase data
in this paper the value $n_{max} = 12$ was used, while $n_{max} = 10$ was used for the particle phase dataset. It is important to note that
the free fitting part does not utilize chemical information in any way. The fitting method is described in the following section
(Sect. 2.1.1). The assignment part of the algorithm is described in detail in Sect. 2.1.3.

### 2.1.1  Peak fitting

Peak fitting is a common process for analyzing mass spectra. It is an attempt to describe the mass spectral signal as a superposition of signals from individual ions, and a background signal. These peaks have a known shape, which is often empirically
determined, and the position of the peak is determined by the mass–to–charge ratio of the ion. In practice these peaks are often
located by a fitting algorithm that minimizes the residual, or unexplained signal. In this work the function used in the fitting
process used for the algorithm is minimizing the $\chi^2$ value given by

$$\chi^2_n = \sum_i^k \frac{(y_i - \hat{y}_{i,n})^2}{\hat{y}_{i,n}}, \tag{1}$$

where $k$ is the number of data points fit to, $y_i$ is datapoint $i$ in the spectrum, and $\hat{y}_i$ is the fit value to this datapoint. Note that
$\hat{y}$ includes both the fit signals and the baseline estimate. Index $n$ denotes the number of peaks included in the fit resulting in
this particular value of $\chi^2_n$. The algorithm is also adapted to be able to fit several spectra simultaneously, for example those
obtained from factorization techniques (Zhang et al., 2019). In this case $\chi^2_{n,tot}$ is calculated as the sum the $\chi^2$ values for each





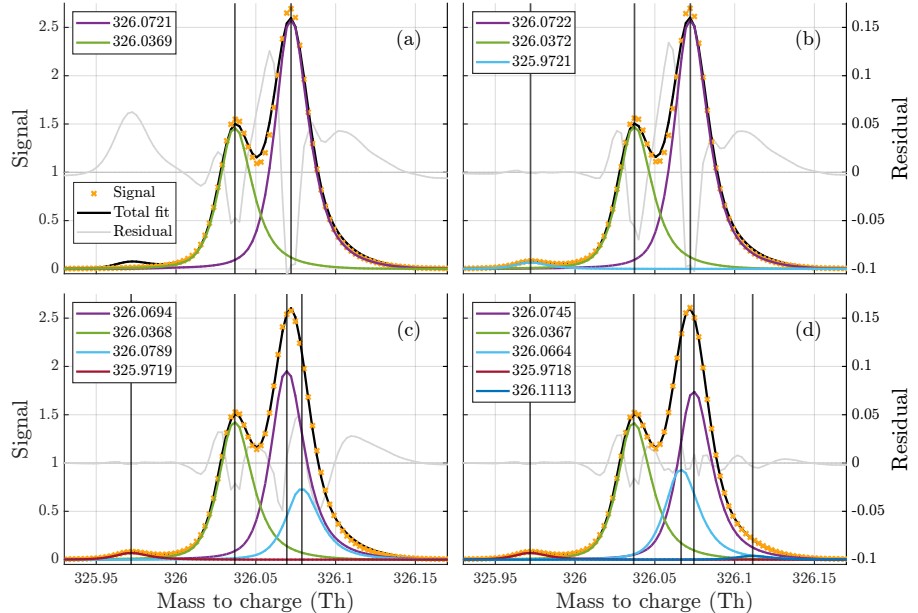

**Figure 1. (a)–(d)** Free fits of 2, 3, 4, and 5 peaks, respectively to the unit mass at 326 Th in the gas phase data set, illustrating the difficulty of the problem. It is quite clear that there are at least three peaks present at the mass. However, the addition of the fourth peak is not as obvious, despite it being assigned far higher signal than the third one included. This illustrates how much the overlap of peaks complicates this problem. It's even less clear if the inclusion of the fifth peak is necessary, and despite it's size it changes the distribution of signal at two of the higher peaks significantly. Both the manual fit, and the algorithm fit to this unit mass are presented later on in Fig. 5.

individual spectrum. In following sections, whenever the peaks are fit, it refers to finding the positions and heights of the peaks that minimize the value of $\chi^2$. Later the $\chi^2$ is also utilized for evaluating the number of peaks that provide the best fit.

Except for the slightly different definition of the minimized function, the algorithm uses the same approach for fitting peaks as Junninen et al. (2010) and therefore the method will not be described here in greater detail. The code used to conduct the peak fitting is also based on the code in the tofTools software developed by Junninen et al. (2010).

### 2.1.2 Determining number of peaks

One of the most difficult problems to solve for the algorithm is to decide what number of peaks to fit for a given unit mass. An
example of some results from free fitting is presented in Fig. 1, the data used for the fit is presented later in Sect. 2.2.2. As can be seen from the figure, the inclusion or exclusion of relatively small peaks, may result in a significant shift in the positions of the peaks contributing the majority of signal at a unit mass. This highlights the importance of choosing the correct number of peaks to fit. To determine the number of peaks fit the following function is used,

$$\text{score}(n) = \text{A} \cdot \frac{\chi_n^2}{\chi_{n_{\max}}^2} + n \cdot \ln(k). \tag{2}$$



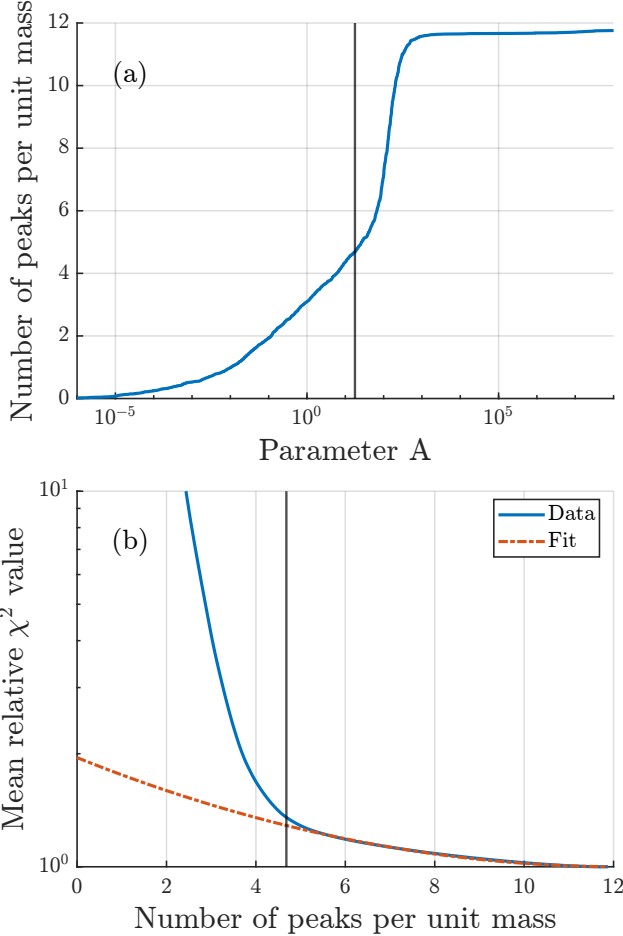

**Figure 2. (a)** How the number of peaks increases with the value of parameter A. As A increases the number of peaks per unit mass approaches the selected value of $n_{max} = 12$. **(b)** How the mean relative goodness of fit score improves with increasing average number of peaks. The fit curve is as described in Sect. 2.1.2. The vertical black line in both figures denotes the algorithm selected value of A.

Here, like before, $k$ and $n$ are the number of data points and the number of peaks, respectively. $\chi^2$ comes from Equation 1, and A is an internal parameter to the algorithm, which will be discussed shortly. Further theoretical motivation of this expression for the score, which is based on the Bayesian information criterion (Neath and Cavanaugh, 2012), is given in the appendix (Sect. A1). However, the main justification for this function is not from theory, but from the results of testing, described in Sect. 2.3 & 3.1. The number of peaks to fit is determined by the $n$–value, that results in the lowest score. However, the final

number of peaks may increase or decrease the number of peaks later, when chemical information is incorporated, as discussed in the following section.

     A somewhat intuitive description of the A parameter is that it determines how much the algorithm values goodness of fit relative to the sparsity of fit peaks. If we select A such that two fits with different number of peaks $n_1$ and $n_2$, have an equal



score we see that A is proportional to the ratio between the difference in peak number ($n$) and the difference in goodness of fit ($\chi^2$):

$$A = \chi^2_{n_{\max}} \cdot \ln(k) \frac{n_2 - n_1}{\chi^2_{n_1} - \chi^2_{n_2}}. \tag{3}$$

This results in a direct relationship between A and the total number of peaks fit by the algorithm $N_{fit}$ (Figure 2a), meaning that determining an optimal value of A is of great importance. Note that A is a single parameter that is used for all unit masses in the entire dataset, however the score function, where A is applied, results in a different number of peaks at each unit mass.

After the free fitting portion of the algorithm is complete, a value of A is defined, which is then used for deciding the number of peaks using the score function (Equation 2). By examining both the synthetic and real datasets analyzed for this paper, the following method for determining a suitable value of A was arrived at. After some number of peaks (roughly 5 peaks per unit mass on average in Fig. 2b), the average relative $\chi^2$ value (i.e. mean($\chi^2_n / \chi^2_{n_{\max}}$)) over all unit masses was roughly proportional to $\exp(-b(\bar{n}_{\max} - \bar{n}(A))^2)$ where $\bar{n}$ is the average number of peaks fit per unit mass. A fit was performed to the higher range of $\bar{n}$, and A was defined as a point where this fit starts to deviate from the data (Figure 2b). A more precise description of the procedure is described in Appendix A3.

### 2.1.3 Assigning formulas

The algorithm iteratively assigns peaks to one integer mass at a time, starting from the lowest mass specified and proceeding to the following integer mass after completing the assignment process at a given mass. The flowchart in Fig. 3 outlines the general approach of the peak assignment. Equations 1 & 2 are both central to this part of the algorithm as well. The general idea behind the structure of this part of the algorithm is to find formulas that match the peaks as well as possible, starting from the most clearly distinguishable peaks. To determine which peak is the easiest to find, the concept of peak significance is used, explained below in Step 2 of the process. The algorithm does however have the option to change a previously assigned peak, in cases where a the presence of a formula becomes much less clear at some later stage in the process (see Step 4 below). In this sense this part of the algorithm mimics the process a human analyst may use when evaluating which formulas are present at an integer mass.

Step 1 of the assignment process is to update the preliminary results using the expected isotopic signals from lower masses. Since the number of peaks may be re-evaluated many times during the assignment process, the algorithm sets the maximum number of peaks that can be fit, $n_{\max}$, at one more than the number of peaks that was determined optimal by the score function, based on the free fits. This saves significant time, and it is also rare that the number of peaks to fit would increases after this point. Often times the number decreases since isotopes from lower masses explain part of the signal. Then the algorithm subtracts the expected isotopic signal from the spectrum, and refits the spectrum at the currently analyzed integer mass (This follows the process described in step 5).

Step 2 evaluates the significance of each peak. This is done by removing the peak, and refitting the heights of the other peaks, while keeping their positions locked. The increase in $\chi^2$ from the fit with all peaks determines the significance of the peak. This difference in $\chi^2$ is then used to ensure that formulas are first assigned to the peaks most important for the overall fit.





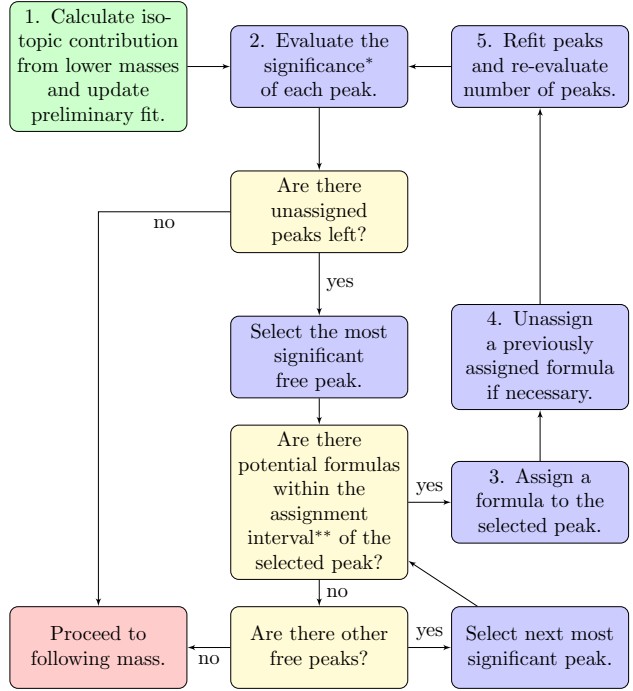

**Figure 3.** Flowchart describing the peak assignment process on a general level. More detailed description of how each step marked with a number works is provided in Sect. 2.1.3. (*) The significance of a peak refers to how much the omission of that peak would increase the $\chi^2$ value of the fit. (**) The assignment interval can be adjusted, but is set to the default value of $0.2\times$FWHM for all of the runs presented in this paper.

In this paper, mentions of the significance of a peak, refer to this difference in $\chi^2$. The use of the word "significance", should not be confused with a strict statistical significance, even though it does serve a similar purpose.

Step 3 Assigns a formula to a selected peak. All the potential formulas are provided to the algorithm as a list, the lists used
for this paper are discussed in Sec. 2.2.4. The selected peak is the most significant peak (according to Step 2), for which a nearby potential formula can be found. This is done by first locating all the potential formulas within the allowed assignment interval, defined as one fifth of the Full Width at Half Maximum (FWHM) of the peak by default. The option resulting in the lowest $\chi^2$ is then assigned to the selected peak. When computing the $\chi^2$ values of the different options, the positions of other peaks are locked in place. Locking the other peaks is to prevent them moving into the position that the selected peak is adjusted
around, but conveniently also saves computing time.

Step 4 is introduced to account for cases where the most recent formula assignment greatly decreases the significance of a peak with a previously assigned formula. A previously assigned formula may be removed if the peak corresponding peak fulfills two conditions. First, the peak with the most recently assigned formula has to have a higher significance than another peak for which for which a formula has been assigned previously (recall that the formulas are assigned in order of descending peak
significance, meaning that the formula assignments must have caused the order to change from the initial situation). Second,




the significance of the other peak must be below 10% (default value) of what it was before the most recent assignment. If these conditions are met for any peak with assigned formula, this formula assignment is removed, and the peak itself appended to the list of free peaks. The list of free peaks is then again sorted based on significance. The algorithm is limited to remove peaks at most ten times, for each integer mass. This is to prevent rare situations where the algorithm ends up in an infinite loop of

assigning and removing peaks. Although removing peak assignments is a relatively common occurrence, reaching the cap of ten removals was very rare, and unlikely in situations where the algorithm was not stuck.

Step 5 re–evaluates the number of free peaks and their positions every time a new formula is assigned. This follows a similar process to the preliminary fits. The lowest number of peaks that may be fit, $n_{\min}$, is the number of peaks that have been assigned a formula, while $n_{\max}$ is adjusted in Step 1. The algorithm then performs fits with numbers of peaks ranging from $n_{\min}$ to $n_{\max}$

peaks, starting from $n_{\min}$. The peaks that have an assigned formula are locked in place, while free peaks are free to change their positions. Each time the number of peaks increments, the position of the added peak is initialized by the most significant remaining free peak in the list of free peaks from the previous step. When the number of peaks to fit exceeds the number of free peaks in the list, the positions are instead initialized using the residual of the fit with one fewer peaks. The fit used for the next iteration of the assignment process is the one resulting in the lowest value of the score function (Equation 2).

At the end of the assignment process there may still be free peaks left that have not been assigned any formula, due to there not being any available options within the assignment interval. These are allowed and are simply labeled "unknown". Although this may be useful for locating some peaks, it should not be relied upon, since isotopes of unknown formulas cannot be accounted for, and may lead to problems at other masses.

If the list of potential elements includes an element whose most common isotope is not the one with lowest mass, the

handling of isotopes mentioned in Step 1 warrants reconsideration. However, atmospheric mass spectra with large numbers of peaks generally consist mostly of formulas made up from the elements C, H, O, and N. The algorithm does not currently check if the isotopes for an assigned formula are present. This may be a useful future improvement, but testing showed it would very rarely be useful in the datasets tested here.

## 2.2 Data sources and description

### 2.2.1 Synthetic data

Synthetic was used frequently during development of the algorithm and for sensitivity test presented in this article (Sect. 3.1). The synthetic spectra were generated as Poisson distributed signals to match the noise that is expected in real mass spectra. This method of data generation is commonly used when attempting to replicate mass spectral signals (Cubison and Jimenez, 2015; Lee and Marshall, 2000). First a noiseless spectrum, $\lambda(m/z)$, was generated as a sum of a constant background level

and signals from individual ions.

$$\lambda(m/z) = \mathrm{BL} + \sum_i^n I_i \cdot f(m/z, \mu_i). \tag{4}$$





Here BL is the baseline signal, $I_i$ is the intensity of the peak indexed $i$, and $f(m/z, \mu_i)$ is the distribution function describing the peak shape, centered at the mass $\mu_i$, accounting for resolution. The peak shape used was empirically estimated from the gas phase data set (Sect. 2.2.2), and the peak locations correspond to real formulas randomly selected from the list of potential formulas for the gas phase data (Sect. 2.2.4). The numbers of peaks per unit mass were uniformly distributed integers between 0 and 8. The generated datasets span the unit mass range from 200–400 Th. The peak intensities were sampled from a lognormal distribution, resulting in a wide variety of intensities.

The noise was included by randomly sampling the signal, $y(m/z)$, at each point from a Poisson distribution with expected value determined by the noiseless $\lambda(m/z)$ (i.e. $y(m/z) \sim \mathrm{Pois}(\lambda(m/z))$). This results in signal dependent noise level, that mimics the expected signal distribution in ToF mass analyzers (Cubison and Jimenez, 2015; Lee and Marshall, 2000). Datasets corresponding to mass spectral resolutions of 4000 and 13000 were generated to mimic the performances of commonly used High resolution ToF (H–ToF) and Long ToF (L–ToF) mass analyzers (Peräkylä et al., 2020).

### 2.2.2 Gas phase data

The gas phase data used in this work has been previously analyzed and published by Peräkylä et al. (2020). This dataset was collected using a nitrate chemical ionization atmospheric pressure interface time of flight mass spectrometer (Jokinen et al., 2012) during chamber experiments of $\alpha$–pinene ozonolysis. The instrument contained a long time-of-flight mass analyzer with a resolving power around 13 000. During the experiment, clean air, $\alpha$–pinene, ozone, and sometimes $NO_2$, water vapor, and inorganic aerosol particles were added to the chamber. The data is only used as a reference of what peaks may be identified during thorough manual analysis of resolution–limited datasets, and therefore the spectra were simply interpolated to a common mass axis, and averaged over the entire measurement period.

### 2.2.3 Particle phase data

The particle phase data was collected using an Aerosol Mass Spectrometer (AMS) (Canagaratna et al., 2007) with an L–ToF mass analyzer at the SMEAR II station in Hyytiälä, Finland, during the spring of 2016, and has previously been analyzed and published by Graeffe et al. (2023). The resolution of the instrument is approximately 5 000 at 100 m/z. The site is surrounded by boreal forest, and the main anthropogenic influence comes from a sawmill about 7 km away. Similarly to the gas phase data, this dataset was averaged over the measurement period to obtain a single average spectrum, to which the algorithm was applied.

### 2.2.4 Formula lists

This work utilized two different lists of potential formulas, one for the gas phase nitrate CIMS dataset, and the other for the particle phase AMS dataset. These lists were generated on broad expectations of what types of ions we expect to detect with the different instruments.



**Table 1.** Quantities used when evaluating algorithm performance.

| Synthetic data | Number | Signal | Real data | Number | Signal |
|---|---|---|---|---|---|
| algorithm fit | $N_{fit}$ | $S_{fit}$ | algorithm fit | $N_{fit}$ | $S_{fit}$ |
| generated data | $N_{gen}$ | $S_{gen}$ | manual fit | $N_{man}$ | $S_{man}$ |
| correctly peaks | $N_{corr}$ | $S_{corr}$ | matching peaks | $N_{match}$ | $S_{match}$ |

The lists of potential formulas provided to the algorithm for the gas phase analysis was generated by providing some combinations of atoms that were sequentially added to form complete molecules. Additional constraints were placed on the constructed molecules by providing limits for the number of atoms of each element, as well as O to C and H to C ratios. For a

complete description of these constraints, see Sect. A2. The gas phase list was also used for both generating and analyzing the synthetic dataset.

The particle phase list was far less constrained, since there are a lot fewer potential formulas at the lower mass range, and the AMS utilizes electron impact ionization, fragmenting the compounds, which means much fewer constraints for ion compositions (Canagaratna et al., 2007). The list is generated mostly by combining atoms of common elements, and does not

include rules motivated by chemistry. For simplicity, only the elements C, H, O, N, and S, were included.

## 2.3 Evaluating algorithm performance

Evaluating the performance of a peak identification algorithm for atmospheric mass spectra is not without its own challenges. Even an experienced analyst cannot be certain about the accuracy of all their fits, and when analyzing mass spectra there is not always a need to attempt to identify all the peaks that may be in the data, since they might not be of relevance or be too

uncertain for further analysis. This is the reason why synthetically generated data was used for most of the development and testing of the algorithm.

For the real data, it is common that there are peaks clearly present in the data, that have not been fitted during manual analysis, either because the peak is not relevant to further analysis, or because it is difficult to find a formula corresponding to the signal. In other instances the manual analysis has included peaks that are more or less clearly not present in the list, as a

part of a series of formulas. Therefore, it is important to remember that the fits used to evaluate the algorithm results are in no way perfect. However, they do represent the information that a typical analyst wants to obtain from the dataset. With this in mind, we use the term *match* rather than *correct*, when referring to a formula that was identified by both the manual analyst and the algorithm. This is to remain conscious of the incompleteness and fallibility of manual identification of peaks.

The quantities used to compare the list provided by the algorithm with either lists of generated peaks or lists provided by

manual analysis are presented in Tab. 1. Here, and in the rest of this article, the letters $N$ and $S$ refer to total number of peaks and total area of signal, while $n$ refers to the number of peaks at one unit mass, and $s$ refers to the area of the signal of one peak. The subscripts *fit*, *gen*, and *corr* refer to *fit by algorithm*, *generated* and *correct* respectively. When discussing the real data, the subscript *gen* is replaced by *man*, to denote *manually fit* instead of *generated*, and the subscript *corr* is replaced by *match*





to denote *a match between manual and algorithm fits*. A peak is considered correct, or matching, if it has the same formula as a peak in the list of generated peaks or manually fit peaks, depending on the data set. $s_{\text{corr}}$ (or $s_{\text{match}}$) is defined as the smaller one of $s_{\text{fit}}$ and $s_{\text{gen}}$ (or $s_{\text{man}}$) for a correctly fit (or matching upon) peak. For an incorrectly fit (or not matching) peak $s_{\text{corr}} = 0$.

### 2.3.1  Synthetic data tests

Four different tests were conducted using synthetic data. The first one investigated how the algorithm performed with different selections of the value for parameter A, evaluated using the metrics from Tab. 1. The results of this test was later used to inform the selection of A explained in Sect. 2.1.2. For this test, and this test only, all the isotopes were removed from the dataset since the inclusion of isotopes makes it difficult to clearly define whether a free peak is in a correct location or not, and this test mostly relied on algorithm results before formula assignment.

The other three tests evaluated how sensitive the algorithm performance was to imprecise inputs from a potential user. The inputs tested were: list of potential formulas, resolution function, and mass calibration. The effect of the potential formula list was tested by adding up to an additional 0–4 molecules of an imaginary element X to each formula in the existing list. Element X had a single isotope with a mass of exactly 1 atomic mass unit, and this addition resulted in many more overlapping formulas for the algorithm to choose from when assigning compositions.

The sensitivity to the resolution function used was tested by applying a resolution scaling factor between 0.9–1.1 to the resolution used in the fit. For example, a factor of 0.9 results in the resolution of the fit peaks is 10% lower than the resolution of the peaks that were generated for the data set, i.e. the algorithm fits peaks that are too wide.

The test for calibration error was done by shifting the generated spectrum by 0–16 ppm before running the algorithm, resulting in a corresponding offset between the peaks in the spectrum and their correct formulas. This results in all the correct formulas being offset by some amount from their actual signal.

## 3  Results

The results of all testing and evaluation of the algorithm are presented in this section. The results are divided into one subsection for each type of data; synthetic, gas phase, and particle phase. The synthetic results focus on the sensitivity tests mentioned in the previous section, and some motivations for the methods the algorithm uses. The results of the application of the algorithm to real data are focused on comparing the algorithm list with the manual list to evaluate how useful the algorithm is as a tool to facilitate that analysis.

### 3.1  Synthetic test results

Synthetic datasets were widely used when developing this algorithm, and many of the methods mentioned earlier, such as the score function (Equation 2), were derived from tests using synthetic data. The method used for selecting the value of the parameter A was mentioned earlier in Sect. 2.1.2. Here we briefly discuss more about the importance of this parameter, and how it impacts the algorithm results. Figure 4a shows how the accuracy of algorithm results change with this parameter. There



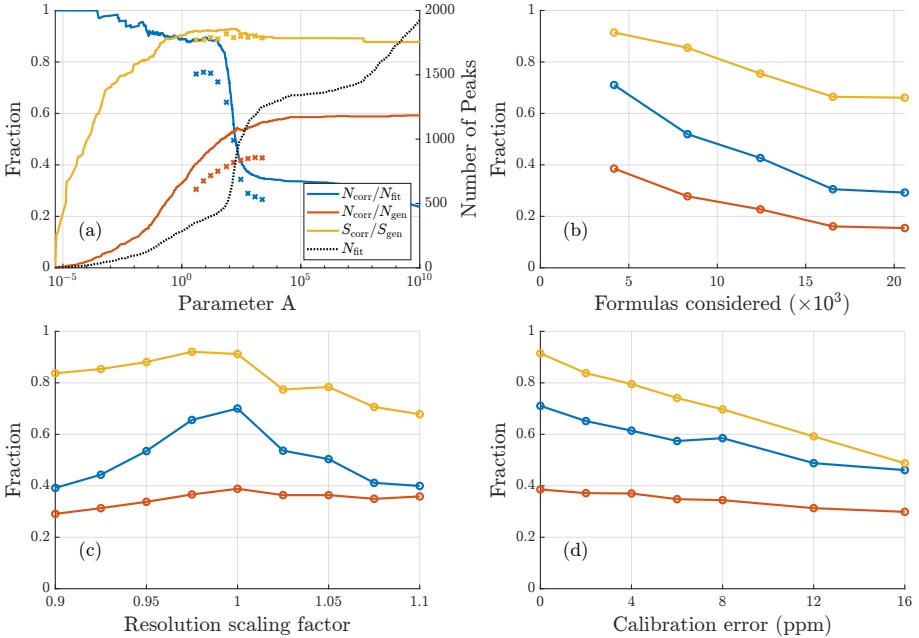

**Figure 4.** Results of sensitivity tests conducted for synthetic data with a resolution around 4000. **(a)** How results vary depending on parameter $A$. The solid lines depict how the scores vary for the free fit, while crosses depict the results after the formula assignment. For the free fits a fit is considered correct if within 0.2·FWHM (50 ppm at 300 Th) of the generated location, after assignment a fit is considered correct only if the precise formula assigned is the same as was generated. Both before and after assignment a sharp drop in the correctly fit fraction of peaks can be seen at around $A = 100$, due to a large number of incorrect peaks being added. **(b)** Influence of the number of compositions in the list of potential formulas. **(c)** Influence of errors in mass calibration. **(d)** Influence of incorrect resolution of fit peaks on fit results. Corresponding results for resolution of 13000 in Fig. A1.

is a critical value of A, at around A=100 in this case, beyond which there is a steep increase in erroneously fit peaks, seen by the sharp increase in total number of peaks, and simultaneous decrease in correctly fit fraction of peaks. Although this point becomes less clear with the inclusion of isotopes and the limited knowledge of fit parameters for real data, the main objective in selecting a value of A is to stay below this critical point. To the left of this point there is an interval of decent options, depending on whether a more conservative (fewer peaks) or exploratory (more peaks) approach is desired. The method for

selecting an appropriate value for A outlined in focuses on landing somewhere fairly close to, but below, the critical value. The mean relative $\chi^2$ value from the critical value of A onward follows the fit curve outlined in Sect. 2.1.2 and displayed in Fig. 2b.

The sensitivity tests for input parameters (Fig. 4b–d) all show expected behavior, with more poorly defined input parameters resulting in worse performance. The test with the expanded list of potential formulas shows the importance of limiting the number of formulas considered. This test was very challenging, with an up to fivefold increase in the number of potential

formulas. The challenge with more potential formulas is not only that there are more options to choose from, but that each erroneously assigned formula leads to the subtraction of expected isotopic signal that is not actually present, while the isotopic





signal of the correct formula remains in the data. This problem can be minimized by limiting the amount of formulas that are considered, or by having higher resolution data (Figure A1b).

Regarding resolution sensitivity, the best results are achieved at the correct resolution of fits, i.e. when the resolution scaling factor is 1. However, there is some asymmetry between lower and higher scaling factors (Figure 4c), with fits using slightly too high resolution resulting in worse results than fits using slightly too low resolution. This may be due to too narrow peaks compensating with additional peaks, which results in erroneous assignments, whereas too wide peaks only limiting assignment of other peaks in instances where there are two neighboring peaks of similar magnitude. However, both of these errors will lead to increased difficulty assigning smaller peaks nearby.

The final sensitivity test addresses mass calibration, and shows that poor mass calibration has a fairly strong impact on the results. Unsurprisingly, a better mass calibration results in better fits. Even small improvements in the mass calibration leads to significantly better fit results, which makes this one of the most important parameters when utilizing the algorithm.

Overall these tests show that optimizing all of these inputs will improve the results of the algorithm. However, realistic accuracy, that can be achieved with currently widely used analysis tools, will not lead to the algorithm being useless. This is further supported by the results of the application of the algorithm to real datasets, where the peak shape and resolution functions were determined empirically, and therefore are as precise as one could expect for a real dataset. These results are presented in the following sections, and show what kind of results or accuracy can be expected from using the algorithm with real data.

## 3.2 Gas phase results

Some example comparisons between the algorithm and manual assignments are presented in Fig. 5. In general, the algorithm is able to adapt the number of peaks required and there is good agreement between the algorithm and the manual fits, at least for the most dominant peaks. These example also show the challenge between a direct comparison with manual analysis. The two formulas, $C_{13}H_{26}O_7N_1^-$ and $C_{12}H_{24}O_9N_1^-$, that contribute close to no signal have probably been included as a part of a series of formulas during manual analysis. Meanwhile, other peaks that were not relevant for the manual analysis due to their negative mass defect were not included in the manual analysis at all. This makes a one to one comparison between algorithm and manual fits misleading, and the following analysis will focus on how much agreement there was between algorithm and manual fits in terms of signal, and the characteristics of the peaks that the algorithm was able, or failed, to find. Formulas, like $C_{13}H_{26}O_7N_1^-$, in the manually compiled list that contributed no signal at all were also omitted from the summary statistics.

Table 2 presents summary statistics of both gas and particle phase results. Overall the algorithm found 76% of all the peaks included in the manual dataset, the found peaks result in a 97% match in assigned signal between the algorithm and the manual analysis.

A more detailed overview of individual peaks and masses is presented in Fig. 6. For each marker in Fig. 6a corresponds to a formula that was identified in the manual analysis, but was not identified by the algorithm. The x–axis value shows how close the closest algorithm fit was to that formula. The red markers represent formulas that were also included in the list of potential formulas provided to the algorithm, black markers represent formulas that were not included in this list. If a formula is included





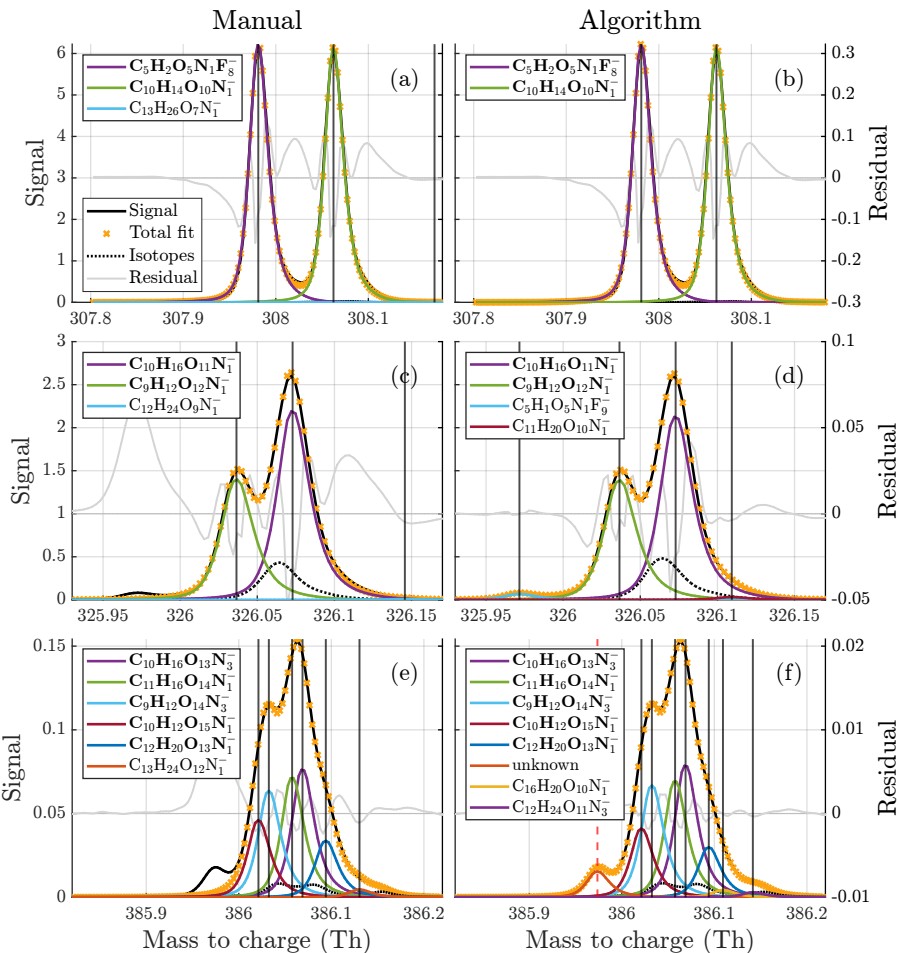

**Figure 5.** Comparisons between manual **(a, c, e)** and algorithm **(b, d, f)** fits to three example unit masses of different complexities. The free fits at 326 m/z was presented earlier in Fig. 1. In addition to the examples here, figures of fits to all of the analyzed integer masses are provided in the supplementary for a better overview of the results.

**Table 2.** Summary statistics for the gas phase and particle phase datasets.

| Dataset | $N_{fit}$ | $N_{man}$ | $N_{corr}$ | $N_{match}/N_{fit}$ | $N_{match}/N_{man}$ | $S_{match}/S_{man}$ |
|---------|-----------|-----------|------------|---------------------|---------------------|---------------------|
| Gas | 1557 | 844 | 644 | 41.4% | 76.3% | 97.1% |
| Particle | 330 | 349 | 241 | 73.0% | 69.1% | 96.8% |

in the list of potential formulas there are three factors influencing how likely it is that the algorithm does not identify a peak in the manual list. The peak may have very low signal compared to other peaks, making it hard to discern, the peak may be





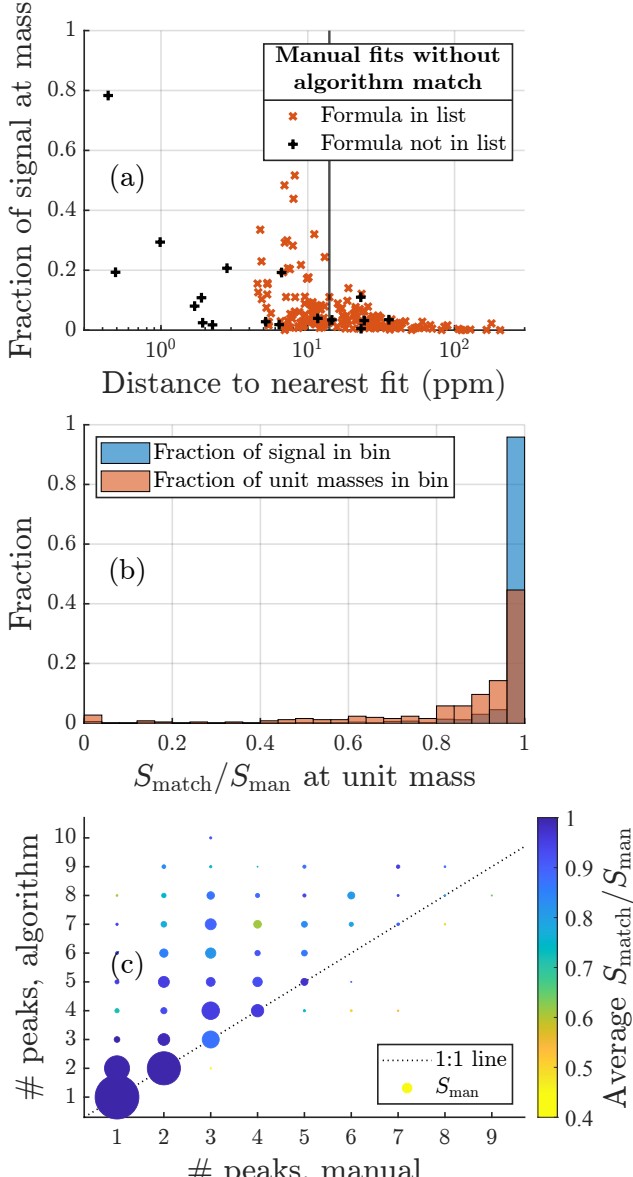

**Figure 6. (a)** Summary of all the formulas that were located in the manual analysis, but not by the algorithm. Red crosses are formulas that were in the list of potential formulas provided to the algorithm. Black markers show formulas that were not on that list. The x–axis is the relative difference in mass between the formula, and the closest peak provided by the algorithm (absolute units of mass in Fig. A4). The y–axis shows what fraction of the total signal area that the formula contributes at its integer mass. The vertical black line marks the assignment interval. **(b)** The distribution of the matching fraction of signal at each unit mass. **(c)** Summary of matching fraction of signal according to the number of peaks fit by each method. Marker area is proportional to the total manually fitted signal.




located very close to another peak, also making it hard to discern, or there is another formula in the list of potential formulas with a very similar mass, resulting in the algorithm misidentifying the peak.

From right to left in the plot, the distance to the nearest fit decreases, and red markers with higher signal start to show up, as the shorter distance between peaks, makes higher relative signals harder to identify, or the algorithm finds signal in the right spot but assigns it the wrong formula. The vertical black line shows the assignment interval. This is the maximum distance allowed between a formula and a given peak, where the formula may be assigned to that peak (see Sect. 2.1.3). For red markers to the left of the vertical black line, there was a free peak in roughly the correct location, but the algorithm decided on another

formula in the list of potential formulas. However, the distance to other nearby peaks, and relative signal intensity, continue to influence how likely misidentification of a peak is. To summarize the meaning of the red markers, the algorithm is most likely to miss peaks that have low signal, that are located close to another peak, or if there are multiple potential formulas within the assignment interval of the peak.

There are also a number of black crosses in Fig. 6a. These represent formulas that were not included in the list of potential

formulas given to the algorithm and therefore could not have been identified by the algorithm. The position of many of these markers in the plot, to the left of the vertical black line, shows that it is in several cases very likely that the algorithm would have located these formulas had they been on the list and a peak in their close proximity was instead labeled as "unknown". This shows that even when some unexpected formulas are present the algorithm may be useful for finding them.

The distribution of the results for individual unit masses are shown in Fig. 6b. Each unit mass is assigned a bin based on the

340 fraction of signal that matched between the manual and algorithm fits ($S_{\mathrm{match}}/S_{\mathrm{man}}$). The y–axis shows the fraction of the total signal area in the spectrum in each bin (blue bars) or the fraction of unit masses in the spectrum in each bin (red bars). First, the algorithm performs exceptionally well for the vast majority of the signal area, with over 97% of the total signal located at masses where more than 94% of the signal matches the manual results. Second, even when looking at just the number of unit masses, the algorithm matches over 80% of manually fitted signal at 80% of unit masses.

Looking at the numbers of peaks fit by the two methods (Figure 6c), it is clear that the algorithm often fits many more peaks than the manual analysis has. However, even when the algorithm does fit many more peaks, it often does not result in poor attribution of signal. Often the higher number of peaks in algorithm fits is due to the manual analysis not attempting to identify every peak. However, there are also a few instances where the algorithm seems to add an excessive number of peaks.

### 3.3 Particle phase results

In general, the algorithm was more restrained in adding peaks when analyzing the particle phase data compared to the gas phase data, and fitted a similar number of peaks to the manual analysis. The reason for this may be related to greater inaccuracies in peak shape and resolution functions. Compared to the gas phase data the resolution and peak shape used, were not as precisely defined, which can be seen from the particle phase fits and residuals (Fig. A8), which may have led to a more conservative fit in general. This resulted in a higher rate of agreement between the manual analysis and the algorithm. In terms of signal the

algorithm achieved a 97% match with the manual analysis also for this dataset. One of the main factors lowering this fraction was the exclusion of trace elements from the list of potential formulas, in the interest of simplicity (Sect. 2.2.4).




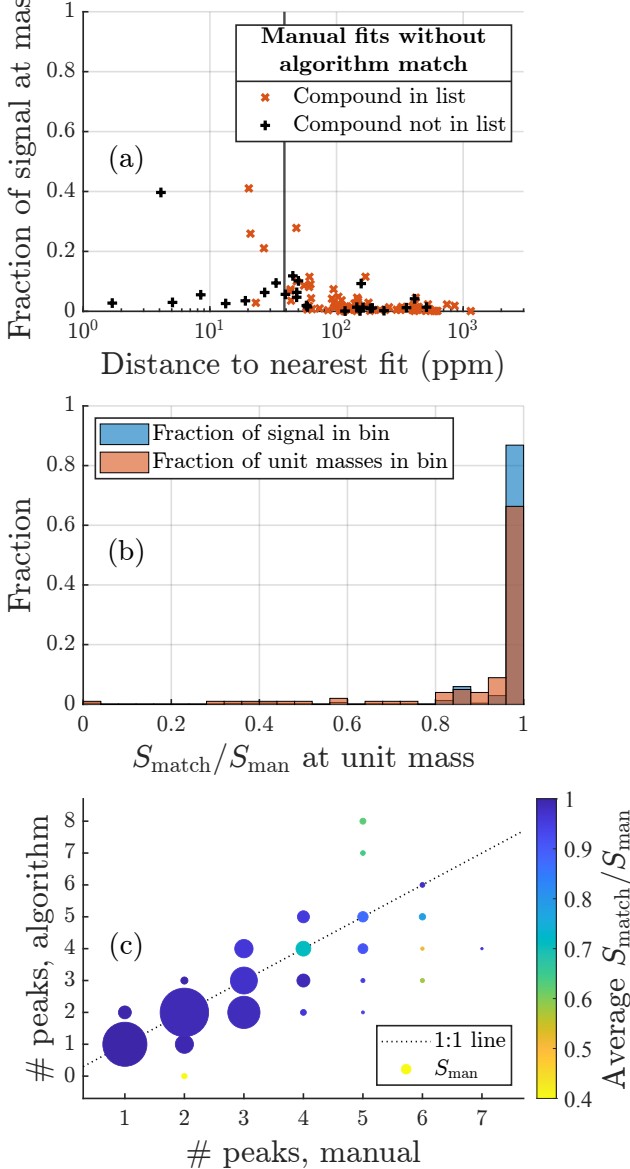

**Figure 7.** Same plots as Figure 6 but for the particle phase data. For this dataset the manual list contained more formulas than the algorithm list. However the general patterns, and overview of the results are very similar.

A more detailed summary of the fits to particles phase data are presented in Fig. 7. In general, the results are very similar to those of the gas phase dataset, which reinforces the points made about the algorithm in the previous section. The one difference is that for this dataset the manual list includes slightly more formulas than the algorithm list. This may be in part due to the more conservative fitting to this dataset mentioned earlier, and in part due to the manual fitting process. For AMS data it is





common to use a long list of compounds and exclude formulas from that list, rather than include them. This may lead to more formulas being used than if starting from an empty list and adding formulas one by one.

## 3.4 Discussion

The results of testing with real datasets show that, despite being a first proof of concept, this algorithm can provide accurate results. The vast majority of the signal is correctly allocated, and a clear majority of the manually identified peaks are also located by the algorithm. These results are very promising, especially considering the strict definition placed on correctly assigned signal used (Sect. 2.3) and the uncertainty present even among manually identified formulas. Although, the algorithm does fit a lot more peaks than the manual analysis for the gas phase dataset, this is often due to the the incompleteness of the manual analysis. Despite there being some occurrences of over fitting by the algorithm, this has not effected the peaks with more significant signal. There is also a natural tool already included in the algorithm to avoid this issue, as a user could adjust parameter A to a lower value to decrease the willingness of the algorithm to fit additional peaks.

Recently, utilizing factorization methods to facilitate peak separation and identification has also been suggested (Zhang et al., 2019). Since the algorithm can fit several spectra at once, this is another area of potential improvement. However, whether the score function can be successfully applied to spectra derived from factor analysis is not certain, since the details of how noise in the data is transferred to the factors are unclear. Therefore, further work would have to be conducted to effectively utilize these two methods together.

The total runtime of the algorithm for the gas and particle phase datasets was around 40 minutes and 5 minutes respectively on a standard laptop. This difference in speed is a combination of the difference in the number of integer masses analyzed, the number of peaks present per integer mass on average, and the set value of $n_{max}$. The algorithm was not optimized for speed, as we believe accuracy is to be prioritized over expediency in this case, as the peak list generation would typically be performed only once for a given dataset. As such, we believe that tens of minutes of additional runtime are worth the investment if it results in far less time ultimately spent to reach the final results.

The sensitivity tests show that fitting parameters also impact algorithm performance. Although the algorithm already achieves good results with fitting parameters of standard accuracy, further improvements in determining these parameters may also result in better algorithm fits. The algorithm results themselves may also be used to better define these parameters, and future works may utilize this for better definitions of peak shape, resolution, and mass calibration functions. Potentially the definition of these can even be improved within the automated process for even better fits.

Another future improvement would be for the algorithm to reconsider formulas, whose isotopic signals do not match the data. As mentioned previously, this was found to be relevant very rarely during testing. In part due to most organics having fairly similar isotopic patterns and in part because the algorithm mostly misidentified peaks with comparatively low signals. Even for datasets containing halogens or other elements with isotopic patterns that deviate from organics, the different mass defect should result in accurate identification of these formulas in a majority of cases. However, this may be an improvement for future consideration.





To summarize, an algorithm was developed with the goal of facilitating the peak fitting and identification process. The
algorithm itself is the main result of this study, and the testing shows that an automated method for identifying peaks in
mass spectra can reach a level of precision that can save users large amounts of time. This new algorithm is completely
automated, and the inputs required are parameters commonly used in analysis of mass spectra. Therefore, the threshold of
adopting this method among users should be low, especially as a part of existing analysis software. Work is currently in
progress to incorporate this process in Tofware (https://www.tofwerk.com/software/tofware/), which is commonly used for
analyzing atmospheric mass spectra. All internal parameters to the algorithm have been determined in a way that a potential
user does not need to worry about adjusting them for their specific dataset, although it is possible for expert users to do so
if desired. The potentially biggest obstacle for adoption of this algorithm is the need for a potential formula list. However,
there are already tools within established analysis software providing lists of formulas containing select elements, that can
be readapted for generating such lists. There have also been calls for better data infrastructure, and databases for a more data
driven approach to the analysis of mass spectra (Sandström et al., 2024). We believe the method described here can complement
this development well; both benefiting from easily accessible lists of formulas, and facilitating the establishment of such lists
as well as reference spectra.

## 4    Conclusions

Mass spectrometers have been a driving force behind recent advances in atmospheric science, and remains a widely utilized
method in the field. However, due to the large number of unique compounds in the atmosphere, peak fitting and identification
from complex mass spectra is a challenging and extremely time–consuming task. Thus, methods that facilitate this task can
save researchers a lot of time.

We have here presented an algorithm for complete automation of the peak fitting and assignment process. The main result of
this work is this algorithm itself. The algorithm was tested on real data from two different mass spectrometers, and the results
show that this algorithm can be a very useful tool for the peak identification process, with a 97% match between manually
identified and algorithmically identified signal intensity for both datasets. The goal of the algorithm is to save time during the
process of analyzing atmospheric mass spectra, and given run times of 40 and 5 min for the gas and particle phase datasets,
respectively, it is clearly much faster for a user to revise the algorithm–generated peak list than to start from scratch.

Sensitivity tests using synthetic data show the importance of correctly defined fitting parameters. Reasonably well defined
parameters do yield good results, as the tests with real data indicate. However, the algorithm results may be significantly
improved by more accurately defined parameters, particularly a good mass calibration. Future work could focus on improving
these parameters within the algorithm itself.

As a proof of concept we believe this work has shown that automated peak fitting and identification can achieve excellent
results. The algorithm described here can already save users a lot of time during peak identification. With further improvements,
and more users providing feedback, the automated fits can likely be improved even further. Work is currently in progress to



include the methods described here into established analysis software, Tofware, which would allow easy utilization of these methods in a wider community.

*Code and data availability.* The algorithm is available as matlab code in its entirety at TBD along with the fit results for all analyzed datasets.

## Appendix A

**A1 Score function**

The Bayesian Information Criterion (BIC) is defined as

$$\text{BIC} = -2\ln\left(L(\Theta_n, \mathbf{y})\right) + n\ln(k). \tag{A1}$$

where $\Theta_n$ is the parameter vector with $n$ parameters, and k is the number of data points in the data matrix $\mathbf{y}$. Assuming every data point is normally distributed around a fit $\hat{\mathbf{y}}_\mathbf{n}$, with variance $\sigma^2$ the likelihood function becomes:

$$L(\Theta_n, \mathbf{y}) = \prod_{i=1}^{k} \frac{1}{\sigma(x_i)\sqrt{2\pi}} \exp\left(-\frac{1}{2}\left(\frac{y_i - \hat{y}_{n,i}}{\sigma(x_i)}\right)^2\right). \tag{A2}$$

And the log-likelihood function is

$$\ln\left(L(\Theta_n, \mathbf{y})\right) = k\ln\left(\frac{1}{\sqrt{2\pi}}\right) + \sum_{i=1}^{k} \ln\left(\frac{1}{\sigma(x_i)}\right) - \frac{1}{2}\sum_{i=1}^{k}\left(\frac{y_i - \hat{y}_{n,i}}{\sigma(x_i)}\right)^2. \tag{A3}$$

Since we are looking for the amount of parameters $k$ that minimizes BIC, the first term is irrelevant, as it is independent of $n$. The second term is also omitted since the change in estimated error should not vary significantly between fits. Counting 440 statistics suggest $\sigma(x) \propto \sqrt{\hat{y}_{n,i}}$. Thus including a proportionality constant, A, yields:

$$\text{BIC} \approx \text{A}\sum_{i=1}^{n}\frac{(y_i - \hat{y}_{n,i})^2}{\hat{y}_{n,i}} + n\ln(k). \tag{A4}$$

Or

$$\text{BIC} \approx \text{A}\chi_n^2 + n\ln(k) \tag{A5}$$

Additional testing of the algorithm showed that the normalization of $\chi^2$ improved results by making the value of A where the 445 numbers of peaks starts to sharply increase be more consistent over a wide range of masses. This finally results in the used score function (Equation 2).

## A2 Generation of potential formula lists

Potential compositions are generated by defining the following parameters:




1. Seeds: The initial parts that a carbon chains are built from. Each composition must contain one and only one seed.

2. Ions : Parts that lead to charging the molecule. Each composition must contain one and only one ion.

3. Parts: The parts that may be added to the carbon chain. Each composition may contain between 0 and some maximum number of each part. Below, when parts are specified the maximum number is before the formula for the part e.g. $5(CH_2)$ means formulas may include between 0 and 5 $CH_2$ units.

Some parts may contain negative numbers of certain elements, such as $-(H^+)$ (deprotonation) or $NO_2(-H)$ (adds $NO_2$ and
removes one H).

**Gas phase formulas**

Additional constraints for gas phase formulas are as follows; all upper/lower limits are inclusive:

- O to C ratio must be between 0.1 and 2.8 (For each nitrogen in the formula, 3 Oxygen are subtracted from the number used to calculate this ratio)

- H to C ratio must be between 0.6 and 2 (Fluorine is counted as H for this ratio)

- Upper limits for atoms of each element: 20C 36H 24O 3N 20F

- lower limits for atoms of each element: 4C 0H 2O 0N 0F

- Formulas without Fluorine with fewer than 4 Oxygen are removed.

- Formulas with exactly 3 Nitrogen and an odd number of Hydrogen are removed.

In the complete list there are three groups of formulas with individually defined parameters:

Group1: Fluorinated carboxylic acids and dicarboxylic acids.

seeds: $CH_2O_2$, $CHO_2F$, $CO_2F_2$

ions: $NO_3^-$, $-(H^+)$

parts: $20(CF_2)$, $1(CO_2)$, $1(O)$, $1(CHF)$

Group2: Closed shell carbon chains/rings.

seeds: $CH_4$, $CH_2O_2$, $C_2H_2O_4$

ions: $NO_3^-$, $-(H^+)$

parts: $20(CH_2)$, $10(CO)$, $3(C)$, $4(O)$, $20(CH_2O)$, $2(NO_2(-H))$, $2(NH)$, $1(HNO_3)$

Group3: Radicals similar to Group2 molecules.

seeds: $CH_3$, $CHO_2$, $C_2HO_4$

ions: $NO_3^-$

parts: $20(CH_2)$, $5(CO)$, $3(C)$, $4(O)$, $20(CH_2O)$



**Particle phase formulas**

Additional constraints for particle phase formulas are as follows; all upper/lower limits are inclusive:

  – O to C ration must be between 0 and 2.

  – H to C ratio must be between 0 and 4. Additionally the number of H atoms must not be less than the number of C atoms $-4$.

  – Upper limits for atoms of each element: 12C 26H 8O 1N 1S.

  – Lower limits for atoms of each element is 0.

In the list there are two groups of formulas with individually defined parameters:

Group1: Compounds without sulfur.

seeds: C

ions: $-e^-$

parts: C, H, O, N

Group2: Compounds with sulfur.

seeds: S

ions: $-e^-$

parts: C, H, O

For group two the upper limits for each element was lowered to 4C 6H 6O and 1S.

**A3   Fitting to select A value**

This section describes the fitting used to select a good value of parameter A, such as the fit shown in Fig. 2b. The fitted function is $\exp(-b(\bar{n}_{\max} - \bar{n}(A))^2)$ as mentioned in Sect. 2.1.2. The fitting is done only within set intervals of $[\bar{n}_{\text{start}}, \bar{n}_{\max}]$, where $\bar{n}_{\text{start}}$ is varied from 0 to $\bar{n}_{\max}$. For the final fit the lowest $\bar{n}_{\text{start}}$ first local minimum of the mean squared error of the fit is selected. If there is no local minimum, the point with the slowest change in mean squared error is selected instead. The purpose of this

selection of $\bar{n}_{\text{start}}$ is to get a fit where the two curves in Fig. 2b match well on the right hand side, without being biased by the left hand side. The point where the mean relative $\chi^2$ value more than 5% higher than the fit value is then selected for parameter A.



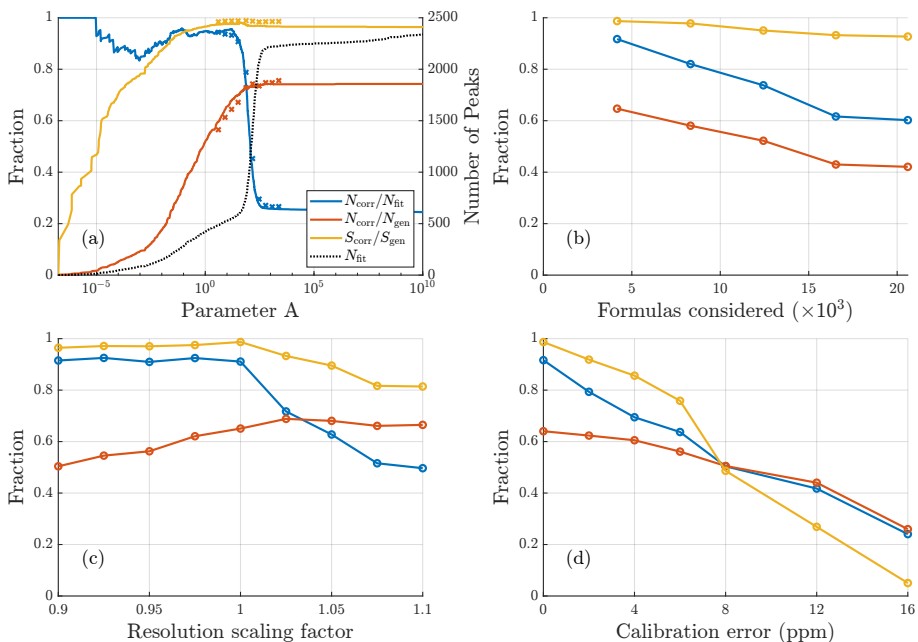

**Figure A1.** Results of sensitivity tests conducted for synthetic data with a resolution around 13000. **(a)** How results vary depending on parameter $A$. The solid lines depict how the scores vary for the free fit, while crosses depict the results after the formula assignment. For the free fits a fit is considered correct if within $0.2 \cdot$ FWHM (14.7 ppm at 300 Th) of the generated location, after assignment a fit is considered correct only if the precise formula assigned is the same as was generated. **(b)** Influence of the number of compositions in the list of potential formulas. **(c)** Influence of errors in mass calibration. **(d)** Influence of incorrect resolution of fit peaks on fit results.



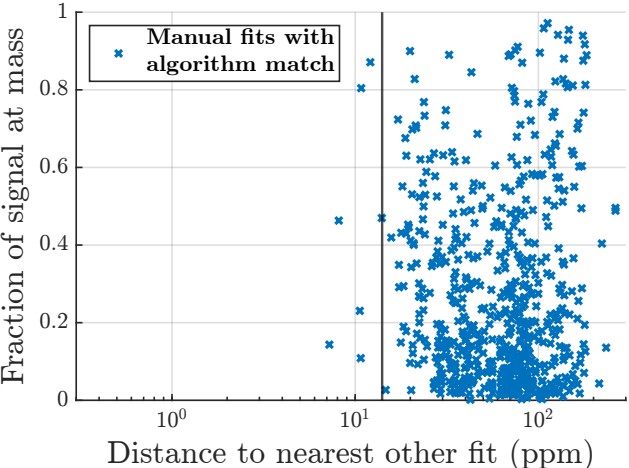

**Figure A2.** Manually identified peaks also identified by the algorithm for the CIMS dataset. The x–axis shows the relative distance to the nearest other peak fit by the algorithm, and the y–axis shows the contribution of that peak to the total signal area at the same unit mass. Note that the closest fit for all of these formulas are actually zero, since the algorithm found the formula in the precise correct location, therefore *other fit* is used for the x–axis label, to denote that it's the closest fit, not corresponding to the correct formula.

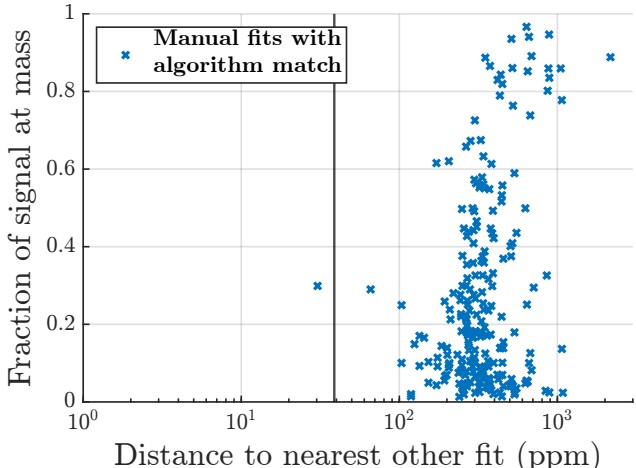

**Figure A3.** Manually identified peaks also identified by the algorithm for the AMS dataset. The x–axis shows the relative distance to the nearest other peak fit by the algorithm, and the y–axis shows the contribution of that peak to the total signal area at the same unit mass. Note that the closest fit for all of these formulas are actually zero, since the algorithm found the formula in the precise correct location, therefore *other fit* is used for the x–axis label, to denote that it's the closest fit, not corresponding to the correct formula.



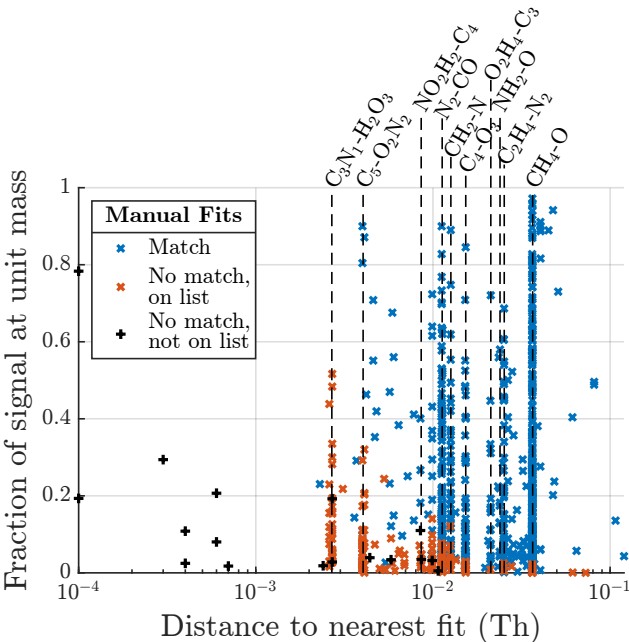

**Figure A4.** Similar to Fig. 6a and Fig. A2, showing all the manually fit formulas to the gas phase dataset, but with absolute units of mass difference. Note that for the blue markers the x–axis distance is to the nearest *other fit*, similarly to Fig. A2. Certain common differences in mass defect are marked with lines in the figure. The lines are labeled according to the two collections of atoms that the difference in mass corresponds to, e.g. $CH_4$–O refers to the difference in mass between $CH_4$ and O (all elemental symbols refer to the most common isotopes of that element). As a further example, if both the manual fit and the algorithm fit identified the two formulas $C_{10}H_{16}O_{13}N_3^-$ and $C_{11}H_{16}O_{14}N_1^-$, then this would result in one blue and one red marker on the dashed vertical line labeled $N_2$–CO, given that there were no other closer peaks. Note that the fact that some mass defects only contain red markers does not suggest that the algorithm always selects the wrong option between the defects, but rather that the algorithm never attempts to fit both of these options. The absence of blue markers is due to that, if the algorithm was to select the correct option, there would not be any other nearby fit, which would move that blue marker somewhere further to the right. Here the black markers are also more clearly separated depending on whether they were labeled as "unknown" or some other formula. If a black marker does not lie on a common mass defect line, it was very likely labeled as "unknown", since the position of the closest fit does not precisely correspond to a common mass difference. Had it been misidentified, the marker would instead likely lie on one of the common mass difference line, since both the algorithm, and manual fits would have assigned it a different chemical formula.





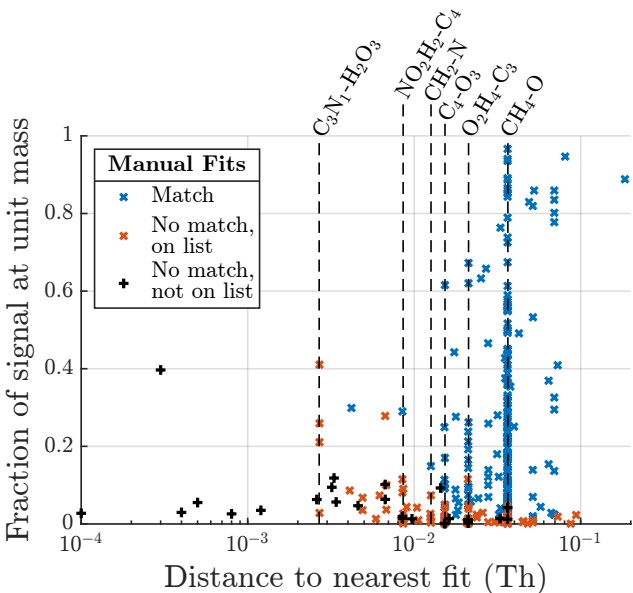

**Figure A5.** Same as previous figure, but for particle phase dataset.

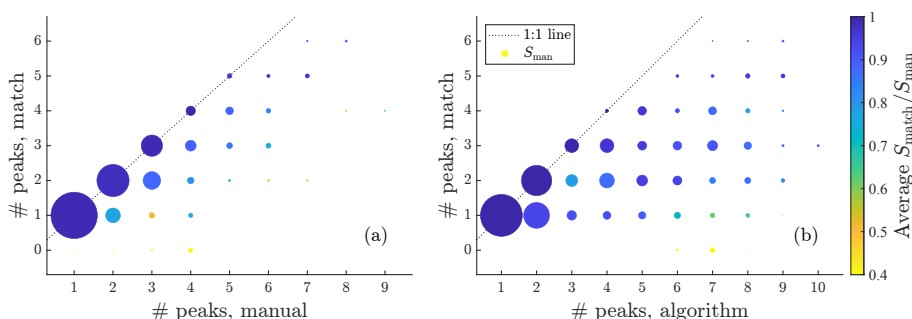

**Figure A6.** Summary of fits to gas phase data in a similar way to 6c. This time instead plotting the number of matching peaks versus the number of manually fit peaks and the number of algorithmically fit peaks, respectively.



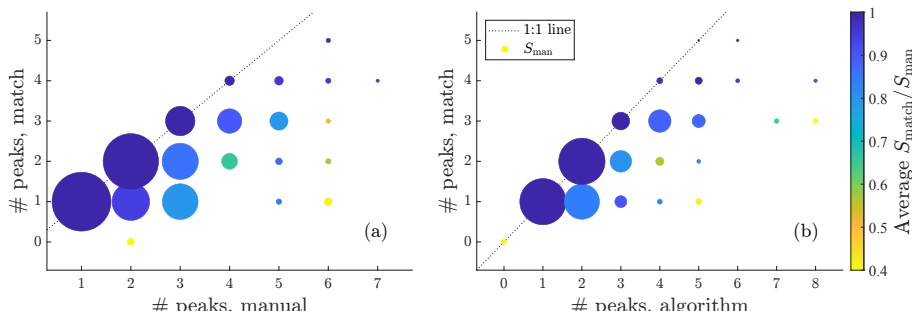

**Figure A7.** Summary of fits to particle phase data in a similar way to Fig. 7c. This time instead plotting the number of matching peaks versus the number of manually fit peaks and the number of algorithmically fit peaks, respectively.

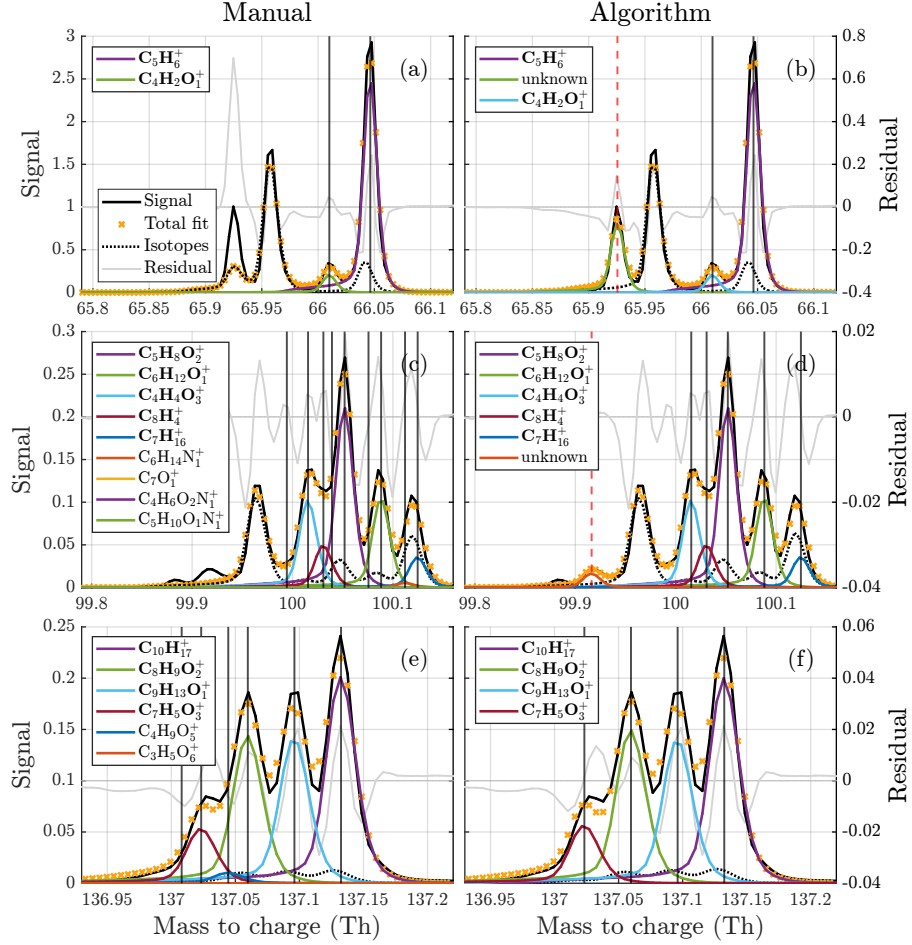

**Figure A8.** Comparisons between manual **(a, c, e)** and algorithm **(b, d, f)** fits to three example unit masses of different complexities for the AMS data.



*Author contributions.* V.M. designed the algorithm, analyzed the results, and prepared the manuscript. M.E., O.P., F.G., and D.W. provided constructive comments on methods and result interpretation. O.P and F.G provided the manual analyses for the gas and particle phase datasets respectively. M.E. assisted in the preparation of the manuscript.

*Competing interests.* The authors declare that they have no conflict of interest.

*Acknowledgements.* This work was supported by Jane and Aatos Erkko Foundation and Svenska Kulturfonden (grants 190437 & 201575). Frans Graeffe also thanks Svenska Kulturfonden for their support (grants 167344 and 177923). We thank the tofTools team for providing tools for mass spectrometry data analysis.



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
