# Peer review of "An Algorithm for Automatic Fitting and Formula Assignment in Atmospheric Mass Spectra"

_EGUsphere, 2024_

## Author Comment (AC1)

Comments in bold font, replies in regular font. Sentences added to the updated manuscript are written in italics.

**RC1, General comment**

**This study presents a technical advancement toward automating peak fitting and formula assignment in mass spectrometric analysis, a task traditionally requiring significant manual effort. The authors developed an algorithm intended to streamline this process by integrating weighted least squares fitting, a modified Bayesian information criterion, and iterative formula assignment. Their approach aims to deliver a preliminary list of likely peaks that can later be refined as needed, thus reducing the time analysts typically spend on labor-intensive, manual identification. The algorithm was tested on gas-phase CIMS and aerosol AMS datasets, and showed comparable accuracy to manual fitting in many cases. However, as with manual methods, lower-intensity signals and interferences from adjacent peaks presented challenges, which led to occasional erroneous assignments. The study's main output — the algorithm — is reported to be undergoing integration into the Tofware analysis software, making it accessible to a broader user base and thus potentially transforming routine data processing workflows. Overall, the work represents a valuable contribution to the field, promising to free up researchers' time for data interpretation rather than time-consuming data processing tasks. I recommend its publication in AMT, considering the following comments are adequately addressed.**

We thank the reviewer for their insights and positive response. The specific comments are addressed below.

**Specific comments**

**Q1. I assume the peak shape function follows a Gaussian distribution? Since position is defined based on the peak shape function, variations in shape could impact the algorithm's reliability.**

It is correct that this definition may impact where the actual fit is made, and it is important that the peak shape used, and it's position, are consistent. The peak shape is a custom function that is extracted from the dataset in established software packages by selecting isolated peaks, normalizing them, and averaging their shapes. The position of the peak is commonly defined as the maximum of the peak shape function. However, as long as the same peak shape is used for the algorithm as was used for the mass calibration, this

should not impact the reliability of the results, since the mass axis will be defined in accordance with the defined position relative to the shape.

We added the following two sentences to clarify this in section 2.1.1: *The position of the peak is commonly defined as the maximum of the fitted peak shape function. It is important that the definition is consistent between mass calibration and the fitted ions to accurately determine the mass–to–charge ratio of the detected ions.*

**Q2. More detail on the rationale for default parameter values, especially for critical values like $n_{max}$ and the parameter A, would be helpful. How robust are these defaults across different instruments and sample types? It would be useful to know how sensitive the fitting results are to these parameter settings and if there is a straightforward way to optimize or validate these parameters for different datasets.**

We agree that straight forward parameters are important for practical use of an algorithm such as this. A user should not need to spend more time on figuring out input parameters than what is saved by adoption of the tool. $n_{max}$ is simply the maximum number of peaks to attempt fitting. We believe someone working on mass spectrometry data will be aware of roughly what number would be suitable here, but the only cost of using a higher number would be longer computation time, so choosing a value a little too high should not be too much of an issue. We expect that the results presented in the manuscript would be quite similar when choosing any value greater than or equal to 10, given that the algorithm never fits more peaks than this. In this work the value 12 is chosen because it would be extremely optimistic to expect anyone to successfully identify 12 peaks at one mass in a spectrum like this. Even a value of 8 should not have a very large impact on the results, since very few masses get assigned that may peaks.

We now clarify the impact of a higher $n_{max}$ value in section 2.1: *Picking a higher number than necessary only costs additional computational resources.*

The parameter A is automatically determined as described on lines 115-121 and in appendix A3. This determination is designed to be as independent of instrument type and dataset as possible. In fact, the very reason A is determined in this manner is to make the algorithm as general as possible. There may still be cases where the method is not perfect, and the algorithm may struggle with certain datasets because of this, since it is difficult to establish a general method without a large variety of datasets. If there are cases where the method is not working in a general manner, it does need to be improved over time. With more users providing feedback improving the generality should be easier.

The same is true for other default values. As a rule we don't believe the default values to be detrimental to algorithm performance in any case, but

some instances may come up where they do require some tweaking. This also is something that requires ample feedback from users, and diverse use cases to identify.

We added clarification on the purpose of the fitting procedure to determine A in section 2.1.2: *The optimal value of this parameter may vary between datasets, so to make the algorithm as general as possible it is automatically determined before the peak identification portion of the algorithm.*

**Q3. The current formula list appears to be derived from existing datasets validated by specific instruments, which are selectively sensitive to certain groups of compounds. While these formulas are relevant to particular compounds, they don't encompass all possible combinations of elements that adhere to established Chemical bonding rules. Given the complexity of organic carbon mixtures in the atmosphere, expanding the list to include additional elements beyond C, H, O, N, and S, and more importantly, to include all possible formula combinations that abbey the valency rules, will be a crucial step for future development.**

Yes, we believe that the potential formula list is very important for practical use of the algorithm. A general list may be useful for newer use cases, but for commonly used mass spectrometers more restrictive lists (such as the ones presented in the manuscript) would likely be more useful. With increased use of this algorithm we hope to provide some type of library where users can find, or even submit, lists for different tools and applications. With the algorithm being able to label peaks "unknown" when there is no suitable compounds, the cost of using a list which does not include all of the detected compound is not so high, and a more restrictive list could avoid having too many formulas with very similar masses.

Choosing the sparsity of the list may also be a useful depending on the goals of the analysis. Perhaps a user is only interested in some class of peaks, and therefore chooses to omit other classes of compounds. For example, when analyzing the gas phase dataset presented in the manuscript it may not be necessary to include fluorinated compounds at all, since they have a very different mass defect from the pure C, H, O, and N containing compounds targeted in the original study.

**Q4. Page 13, line 301: Following the previous comment, implementing the odd nitrogen rule would serve as a valuable criterion for automatically excluding incorrect formulas as discussed here.**

We agree that the odd nitrogen rule may be useful for many datasets. This is the reason why "Formulas with exactly 3 Nitrogen and an odd number of Hydrogen" are not included in the gas phase list (note that one nitrogen is expected to come from the adduct ion). However, the nitrogen rule does not apply for radicals, and we do actually expect to see some of those in the gas

phase dataset. This is why the nitrogen rule is only enforced for the "rarer" 3 nitrogen formulas.

We now clarified our reasoning in Appendix A2: *The last criterion on the list is to make sure the formulas with three nitrogen follow the nitrogen rule. The nitrogen rule is not considered for other numbers of nitrogen atoms since formulas with 0–2 nitrogen are more abundant, and even radicals can be observed.*

In the particle phase data we detect mostly fragments of molecules that have been vaporized. Therefore we cannot restrict the data very much based on chemistry for that dataset.

We also added clarification on the decisions for generating the particle phase dataset: *Both the constraints and the parts used to build the formulas in this list are less general than for the gas phase dataset. This is because the AMS mostly detects fragments of molecules so valency rules cannot be used to constrain the list.*

**Q5. Given that low-intensity peaks are prone to interference and misassignment, further explanation on strategies for handling such peaks would strengthen the method. A more rigorous treatment of noise and background signals, including methods for background subtraction, and options to customize baseline input, could improve accuracy.**

The second referee asked a similar question and this reply aims to answer both of these questions. The same reply is given in response to Referee 2 Question 16.

As mentioned in the comment, background is important, but isolated low-intensity peaks are not that challenging. The real problems are the low intensity peaks overlapping with other (potentially also low intensity) peaks. Here the main problem in our experience is the compounding errors from mass calibration and peak shape from the other nearby peaks. These errors will add up from all the nearby peaks, and additionally a small error in a big peak may have a huge impact on a small peak. In addition, there is the noise from counting statistics contributed by the overlapping peaks, further decreasing the signal of the low-intensity peak relative to the noise.

For most of these issues the solution is to better define the parameters i.e. minimize calibration errors, and errors in peak shape. Also removing as many unrealistic compounds as possible from the potential list may be helpful, but this runs the risk of missing something of scientific importance. Improving the accuracy of these parameters is not straight forward. Some of it relies on the expertise of the analyst, and their familiarity with the dataset, and some on the available tools. With user friendly tools, even an inexperienced analyst could achieve good results. We have not discussed these parameters extensively in the manuscript since they are treated as input parameters for

the algorithm. However, the two don't have to be treated separately forever. Although outside the scope of this work, including some refining of these parameters in the algorithm workflow could lead to significant improvements in results. This would be the next step in improving this algorithm further.

It is also important to note that this is not an algorithm specific problem. There is a limit of what can be inferred from the provided dataset due to the signal–to–noise ratio, and unavoidable uncertainty in parameters, such as mass calibration. Regardless of if the fitting is done manually or automatically, there will be some signals that cannot be assigned with confidence, or cannot be distinguished from the noise at all.

Concerning the baseline in particular, the algorithm seems to handle it quite well. A baseline is fit at each mass, as a constant background. Fitting the baseline along with the peaks is important, because the algorithm will otherwise be highly sensitive to the input baseline, and small errors will result in very big issues with the low- or even medium-intensity signals. However, baselines are not always constant, so the option to fit a linear, or even quadratic baseline could improve this approach further. Currently, non-constant baselines can be handled by the user inputting a baseline, and the algorithm adding a fitted baseline on top of that.

The following was added to the Discussion (section 3.4): *As evidenced by the results, the most challenging problem with peak identification are the peaks with lower signal, overlapped by other peaks. These peaks are more significantly impacted by errors in mass calibration, peak shape, or assignments since the errors of overlapping peaks add up. The difference in signal also means that a small inaccuracy in the peak shape for a large peak can lead to a large impact on a smaller peak in the vicinity. Therefore we believe the primary means of improving the results in general, but most significantly for the lower intensity signals, is to improve the methods to accurately determine these fitting parameters.*

We also added the following paragraph a bit later on in the same section: *However, since peak fitting is a statistical tool, there will always be an inherent level of uncertainty. Whether the identification is done manually or by an algorithm, there will be some peaks that cannot be identified with the desired confidence. Where this limit is encountered depends on the input data. The goal of the algorithm is to save time, so we do not think it is necessary to demand it to be able to identify more peaks than manual fitting.*

**Q6. I am not sure if isotopic checks are currently incorporated in the algorithm, but incorporating an optional check for isotopes could reduce misassignments, particularly for elements with non-standard isotopic distributions**

We did consider this as a potential tool, and we did some tests for this. The topic of isotopic checks is discussed like this in the manuscript:

(Section 2.1): "The algorithm does not currently check if the isotopes for an assigned formula are present. This may be a useful future improvement, but testing showed it would very rarely be useful in the datasets tested here."

To clarify, we have only found one single case in the results where an isotope check would have changed a peak assignment.

(Section 3.4): "Another future improvement would be for the algorithm to reconsider formulas, whose isotopic signals do not match the data. As mentioned previously, this was found to be relevant very rarely during testing. In part due to most organics having fairly similar isotopic patterns and in part because the algorithm mostly misidentified peaks with comparatively low signals. Even for datasets containing halogens or other elements with isotopic patterns that deviate from organics, the different mass defect should result in accurate identification of these formulas in a majority of cases. However, this may be an improvement for future consideration."

We hope this adequately addresses the concerns of the referee.

---

## Author Comment (AC2)

Comments in bold font, replies in regular font. Sentences added to the updated manuscript are written in italics.

**RC2, General comment**

**This study introduces an automated algorithm to streamline peak fitting and formula assignment in mass spectrometry for atmospheric analysis. The algorithm uses weighted least squares fitting and a modified Bayesian information criterion to identify peaks in mass spectra. It was tested on synthetic data and real datasets from gas-phase oxidation using chemical ionization mass spectrometry and particle measurement by aerosol mass spectrometry, yielding results comparable to manual methods but much faster. Errors were mainly observed with low-intensity signals affected by higher-intensity interference. Despite these errors, the algorithm offers a valuable starting point for peak identification and can be manually refined if necessary. Overall, the manuscript is well written. The technique is useful and valuable to the community.**

We thank the reviewer for their insights and positive response. The specific comments are addressed below.

**Specific comments**

**Q1. Line 144: "assigns," not "Assigns."**
Changed according to the referees suggestion.
**Q2. Line 154: Two "for which" are redundant.**
Changed according to the referees suggestion. We also noticed an additional "peak" on line 152 which we removed.
**Q3. Why was a default value of $0.2 \times$ FWHM used? Have you performed a sensitivity analysis to determine this value?**
Since the algorithm evaluates all the options within the assignment interval separately, the interval of $0.2 \times$ FWHM is mostly relevant for defining when a peak is far enough from any candidate compositions to define it as "unknown". It is not selected based on testing, but based on the idea that it is unreasonable to expect the correct formula to be further away from the fit than this distance, even if there is no other potential formula present.

This value may be important to tune when the list of candidate compounds is lacking, and may depend on other parameters such as signal to noise ratio, or calibration errors. However, we do not believe it will have a very big impact on results presented here.

We added the following clarification to Section 2.1.3: *The default assignment interval of $0.2 \times FWHM$ represents the minimum distance between a peak*

*and any potential formula for the peak to be labelled "unknown". Since all the potential formulas that are within the interval are tested by fitting, the value is primarily selected to be wide enough not to exclude the correct formula. Tweaking this value may be relevant if the list of potential compounds is more or less restrictive.*

**Q4. Line 156: The term "the other peak" is confusing. Is it "the peak" or "another peak" in line 154?**

Yes, "The other peak" was supposed to refer to "another peak". We agree that this was confusing and modified the section as follows: *First, the peak with a previously assigned formula must have lower significance than the peak with the most recently assigned formula (recall that the formulas are assigned in order of descending peak significance, meaning that the formula assignments must have caused the order to change from the initial situation). Second, the significance of the peak with a previously assigned formula must be below 10% (default value) of what it was before the most recent assignment.*

**Q5 & Q6. Line 132: How is the isotopic contribution calculated without assigning chemical formulas first? Line 162: Should step 5 go back to step 1 since changes in the number of peaks and chemical formulas also affect the isotopic contribution?**

Since the most common isotopes of C, H, O, N, S are all lighter than the rarer isotopes, and F only has one naturally occurring isotope, all the calculated isotopic signals will be found at higher integer masses. Since the algorithm analyzes one integer mass at a time, starting from the lowest mass, all of the isotopes for the formulas identified at the currently processed integer mass will only affect later masses and the isotopic signal does not need to be updated until the algorithm proceeds to the next integer mass. If elements that do not follow this rule, such as Fe, it is best to fit the naturally occurring isotope with the lowest mass, rather than the most common one.

When the algorithm proceeds to a new integer mass the ions giving rise to the isotopic signals have already been assigned at the previous masses. Therefore, the isotopic signals at the current mass can be calculated before assigning any formulas. The exception is the lowest (first) mass where no isotopic contribution can be accounted for.

We clarified in the caption of Fig. 3 that the flowchart represents the process at a single integer mass. We also described in greater detail the reasoning behind how isotopic signals are handled in section 2.1.3 (Step 1): *The isotopic signal is calculated from the expected isotopic ratios of each formula at lower integer masses. For the common elements detected in the ambient considered here, the rarer isotopes all have higher mass than the most common one, so isotopes from ions detected at the current integer mass or higher don't need to be considered. For elements where this is not the case, e.g. iron, the lowest mass isotope can be fit rather than the most abundant one.*

**Q7. Line 180: Please provide more details on how the synthetic data was generated and the mass spectrum for better understanding and visualization.**

We added the following figure to the appendix to visualize the data generation process. Some additional details are provided in the caption.

[Figure]

**Figure 1:** *Example of synthetic data generation at integer mass 306 with a mass resolution of 4000. Three ions, with exact positions displayed in the legend, along with the baseline contribute to the expected signal $\lambda$. Each point in the generated spectrum, $y$, is sampled from a Poisson distribution with expected value $\lambda$, resulting in noise with a standard deviation of $\sqrt{\lambda}$.*

**Q8. Line 187: How was the baseline determined?**

The purpose of the synthetic data was to test how the algorithm results respond to suboptimal input parameters, and get an idea of how the number of peaks can be determined in a general way. Therefore, the baseline and mean signal intensity of generated signals were chosen such that the algorithm would be challenged. As can be seen from Fig. 4 only a little over 70% of the formulas identified for the synthetic data were correct, despite perfect knowledge of peak shape and resolution functions. This was due to the generated data being quite noisy by design, to challenge the algorithm. Because the main purpose of the synthetic data was to evaluate the response to imperfect inputs, the baseline was selected by roughly estimating a reasonable average ratio of background to peak heights from real data. Due to the peak intensities being sampled from a lognormal distribution this ratio varies over several orders of magnitude, similarly to real mass spectra.

We added the following explanation to section 2.2.1 to describe the variety in both signal intensity and signal–to–baseline ratio: *The total intensity allocated to all peaks at an integer mass was sampled from a lognormal distribution. That signal was then distributed between peaks at the same unit mass*

*by weights sampled from a uniform distribution. This results in high variety in intensity between integer masses, and high variety of signal–to–background ratios, while restricting the number of unit masses where a single peak contributes nearly all of the signal.*

**Q9. Lines 203-205: Why is fluorine included for generating gas-phase formulas for alpha-pinene ozonolysis products?**

Some fluorinated compounds are emitted from the teflon tubing, chamber and fan used in the experimental setup. Such fluorinated contaminants are commonly observed with most negative ion CIMS instruments, such as those using $NO_3^-$ or $I^-$ (e.g. https://doi.org/10.1021/acs.est.3c09255, https://doi.org/10.5194/acp-20-5945-2020, https://doi.org/10.5194/acp-12-5113-2012).

We added a description of this in the appendix A2: *The Fluorine containing compounds are commonly observed in datasets from negative ion CIMS instruments and originate from the teflon tubing, chamber and fan used in the experimental setup (Mattila et al. 2024, Zhang et al. 2020, Ehn et al. 2012).*

**Q10. Line 261: What is meant by "calibration error"? Does it occur during the calibration process before peak assignment?**

Calibration error refers to a shift in the mass axis of the spectrum due to imperfect mass calibration of the acquired data.

We added the following clarification in section 2.3.1: *Calibration error refers to an error in the definition of the mass axis of the mass spectrum. This results in the signals from all ions being offset from their actual mass in the spectrum.*

**Q11. Figure 4: For $N_{corr}$ and $S_{corr}$, what do you mean by "correct"? How do you know they are correct? Is it because they are based on synthetic data?**

Yes, these tests were conducted for synthetic data, so all the peaks that are present are known. The caption states a more clear definition of what is considered correct in the A panel, where the peaks depicted by the solid lines have not yet been assigned a formula: "For the free fits a fit is considered correct if within $0.2 \times$FWHM (50 ppm at 300 Th) of the generated location, after assignment a fit is considered correct only if the precise formula assigned is the same as was generated".

We clarified that the purpose of the synthetic dataset is to know the correct fits in section 2.3: *This is the reason why synthetically generated data for which all the correct peak positions and signal intensities are known was used for most of the development and testing of the algorithm.*

**Q12. Lines 320-325: To help readers, provide the peak lists for the gas-phase and particle-phase tests based on the rules in Appendix A2.**

The gas and particle phase lists contain 11096 and 1515 formulas respectively. We believe this is too many to print entirely in the manuscript. Instead

we added tables A1 and A2 to the appendix, showing the potential formulas for the integer masses shown in the example plots, to give a sense of the acceptable compounds. The data for the full lists will be made available along with the algorithm code.

**Q13. Lines 342-344: It's unclear how the values of 97%, 94%, and 80% were determined. Please clarify.**

These values are read from Figure 6b. We adjusted the manuscript to clearly reference where the information on the mentioned lines is from.

**Q14. Line 352: Should peak shape be considered before step 1?**

Peak shape is something we believe should always be optimized when analyzing complex mass spectra, and there are pretty good tools already available for this. Since there are tools for this, we consider the peak shape to be an input parameter, and the peak shape function is considered at every step of the algorithm. The algorithm does not currently attempt to change or optimize it in any way, since it is outside the scope of this study.

Since our aim was to demonstrate the performance of the algorithm we do not consider it a weakness of the testing data that one of the datasets does not have the best defined peak shape, since this shows that the algorithm works even in non-ideal situations, therefore we did not attempt to improve the input data in any way.

**Q15. Lines 394-407: Consider combining this paragraph into the Conclusion section.**

We completely removed the summary introducing the paragraph, as that information was already clearly stated in the conclusion. However, we believe the following discussion about practical adoption of the algorithm suits the discussion section better, so we did not move it.

**Q16. The authors mentioned that errors were mainly observed with low-intensity signals. Are there strategies to minimize errors in peak assignment for low-intensity peaks? Please discuss.**

The first referee asked a similar question and this reply aims to answer both questions. The same reply is given in response to Referee 1 Question 5.

As mentioned in the fifth comment by Referee 1, background is important, but isolated low-intensity peaks are not that challenging. The real problems are the low intensity peaks overlapping with other (potentially also low intensity) peaks. Here the main problem in our experience is the compounding errors from mass calibration and peak shape from the other nearby peaks. These errors will add up from all the nearby peaks, and additionally a small error in a big peak may have a huge impact on a small peak. In addition, there is the noise from counting statistics contributed by the overlapping peaks, further decreasing the signal of the low-intensity peak relative to the noise.

For most of these issues the solution is to better define the parameters i.e. minimize calibration errors, and errors in peak shape. Also removing as many

unrealistic compounds as possible from the potential list may be helpful, but this runs the risk of missing something of scientific importance. Improving the accuracy of these parameters is not straight forward. Some of it relies on the expertise of the analyst, and their familiarity with the dataset, and some on the available tools. With user friendly tools, even an inexperienced analyst could achieve good results. We have not discussed these parameters extensively in the manuscript since they are treated as input parameters for the algorithm. However, the two don't have to be treated separately forever. Although outside the scope of this work, including some refining of these parameters in the algorithm workflow could lead to significant improvements in results. This would be the next step in improving this algorithm further.

It is also important to note that this is not an algorithm specific problem. There is a limit of what can be inferred from the provided dataset due to the signal–to–noise ratio, and unavoidable uncertainty in parameters, such as mass calibration. Regardless of if the fitting is done manually or automatically, there will be some signals that cannot be assigned with confidence, or cannot be distinguished from the noise at all.

Concerning the baseline in particular, the algorithm seems to handle it quite well. A baseline is fit at each mass, as a constant background. Fitting the baseline along with the peaks is important, because the algorithm will otherwise be highly sensitive to the input baseline, and small errors will result in very big issues with the low- or even medium-intensity signals. However, baselines are not always constant, so the option to fit a linear, or even quadratic baseline could improve this approach further. Currently, non-constant baselines can be handled by the user inputting a baseline, and the algorithm adding a fitted baseline on top of that.

The following was added to the Discussion (section 3.4): *As evidenced by the results, the most challenging problem with peak identification are the peaks with lower signal, overlapped by other peaks. These peaks are more significantly impacted by errors in mass calibration, peak shape, or assignments since the errors of overlapping peaks add up. The difference in signal also means that a small inaccuracy in the peak shape for a large peak can lead to a large impact on a smaller peak in the vicinity. Therefore we believe the primary means of improving the results in general, but most significantly for the lower intensity signals, is to improve the methods to accurately determine these fitting parameters.*

We also added the following paragraph a bit later on in the same section: *However, since peak fitting is a statistical tool, there will always be an inherent level of uncertainty. Whether the identification is done manually or by an algorithm, there will be some peaks that cannot be identified with the desired confidence. Where this limit is encountered depends on the input data. The goal of the algorithm is to save time, so we do not think it is necessary to*

*demand it to be able to identify more peaks than manual fitting.*

**Q17. Have the authors compared this algorithm with others mentioned in lines 42-43? What are its advantages and limitations?**

Sandström et al. describe the need for mass spectral databases, which could potentially be used to bypass peak assignment in certain situations, but the paper just discusses the potential of the approach and does not provide anything practically useful at the moment. Alton et al. present a generalized Kendrick analysis method as a visualization tool, which may facilitate the formula assignment process. Zhang et al. present a factorization method which may be useful in achieving better separation between overlapping peaks in mass spectra.

None of the above mentioned methods provide an actual list of compounds, and therefore cannot be directly compared to the algorithm presented here. However, as a future work they may be useful to incorporate into the algorithm.

Stark et al. does present an algorithm, not that different from the one described here. However, the method is only used to provide bulk chemical information about the sample, such as carbon to oxygen ratios, or average oxidation state. Due to the very different goals of these algorithms we have not made a direct comparison of the results.